# Enforcing balance allows local supervised learning in spiking recurrent networks

**Ralph Bourdoukan**
Group For Neural Theory, ENS Paris
Rue dUlm, 29, Paris, France
ralph.bourdoukan@gmail.com

**Sophie Deneve**
Group For Neural Theory, ENS Paris
Rue dUlm, 29, Paris, France
sophie.deneve@ens.fr

## Abstract

To predict sensory inputs or control motor trajectories, the brain must constantly learn temporal dynamics based on error feedback. However, it remains unclear how such supervised learning is implemented in biological neural networks. Learning in recurrent spiking networks is notoriously difficult because local changes in connectivity may have an unpredictable effect on the global dynamics. The most commonly used learning rules, such as temporal back-propagation, are not local and thus not biologically plausible. Furthermore, reproducing the Poisson-like statistics of neural responses requires the use of networks with balanced excitation and inhibition. Such balance is easily destroyed during learning. Using a top-down approach, we show how networks of integrate-and-fire neurons can learn arbitrary linear dynamical systems by feeding back their error as a feed-forward input. The network uses two types of recurrent connections: fast and slow. The fast connections learn to balance excitation and inhibition using a voltage-based plasticity rule. The slow connections are trained to minimize the error feedback using a current-based Hebbian learning rule. Importantly, the balance maintained by fast connections is crucial to ensure that global error signals are available locally in each neuron, in turn resulting in a local learning rule for the slow connections. This demonstrates that spiking networks can learn complex dynamics using purely local learning rules, using E/I balance as the key rather than an additional constraint. The resulting network implements a given function within the predictive coding scheme, with minimal dimensions and activity.

The brain constantly predicts relevant sensory inputs or motor trajectories. For example, there is evidence that neural circuits mimic the dynamics of motor effectors using internal models [1]. If the dynamics of the predicted sensory and motor variables change in time, these models may become false [2] and therefore need to be readjusted through learning based on error feedback.

From a modeling perspective, supervised learning in recurrent networks faces many challenges. Earlier models have succeeded in learning useful functions at the cost of non local learning rules that are biologically implausible [3, 4]. More recent models based on reservoir computing [5–7] transfer the learning from the recurrent network (with now "random", fixed weights) to the readout weights. Using this simple scheme, the network can learn to generate complex patterns. However, the majority of these models use abstract rate units and are yet to be translated into more realistic spiking networks. Moreover, to provide a sufficiently large reservoir, the recurrent network needs to be large, balanced and have a rich and high dimensional dynamics. This typically generates far more activity than strictly required, a redundancy that can be seen as inefficient.

On the other hand, supervised learning models involving spiking neurons have essentially concentrated on the learning of precise spike sequences [8–10]. With some exceptions [10,11] these models use feed-forward architectures [12]. In a balanced recurrent network with asynchronous, irregular and highly variable spike trains, such as those found in cortex, the activity has been shown to be

chaotic [13, 14]. This leads to spike timing being intrinsically unreliable, rendering a representation of the trajectory by precise spike sequences problematic. Moreover, many configurations of spike times may achieve the same goal [15].

Here we derive two local learning rules that drive a network of leaky integrate-and-fire (LIF) neurons into implementing a desired linear dynamical system. The network is trained to minimize the objective $\|\mathbf{x}(t) - \hat{\mathbf{x}}(t)\|^2 + H(\mathbf{r})$, Where $\hat{\mathbf{x}}(t)$ is the output of the network decoded from the spikes, $\mathbf{x}(t)$ is the desired output, and $H(\mathbf{r})$ is a cost associated with firing (penalizing unnecessary activity, and thus enforcing efficiency). The dynamical system is linear, $\dot{\mathbf{x}} = \mathbf{Ax} + \mathbf{c}$, with $\mathbf{A}$ being a constant matrix and $\mathbf{c}$ a time varying command signal. We first study the learning of an autoencoder, i.e., a network where the desired output is fed to the network as a feedforward input. The autoencoder learns to represent its inputs as precisely as possible in an unsupervised fashion. After learning, each unit represents the encoding error made by the entire network. We then show that the network can learn more complex computations if slower recurrent connections are added to the autoencoder. Thus, it receives the command $\mathbf{c}$ along with an error signal and learns to generate the output $\hat{\mathbf{x}}$ with the desired temporal dynamics. Despite the spike-based nature of the representation and of the plasticity rules, the learning does not enforce precise spike timing trajectories but, on the contrary, enforces irregular and highly variable spike trains.

## 1   Learning a balance : global becomes local

Using a predictive coding strategy [15–17], we build a network that learns to efficiently represent its inputs while expending the least amount of spikes. To introduce the learning rules and explain how they work, we start by describing the optimized network (after learning).

Let us first consider a set of unconnected integrate-and-fire neurons receiving shared input signals $\mathbf{x} = (x_i)$ through feedforward connections $\mathbf{F} = (F_{ji})$. We assume that the network performs predictive coding, i.e. it subtracts from each of these input signals an estimate $\hat{\mathbf{x}}$ obtained by decoding the output spike trains (fig 1A). Specifically, $\hat{x}_i = \sum D_{ij} r_j$, where $\mathbf{D} = (D_{ij})$ are the decoding weights and $\mathbf{r} = (r_j)$ are the filtered spike trains which obey $\dot{r}_j = -\lambda r_j + o_j$ with $o_j(t) = \sum_k \delta(t - t_j^k)$ being the spike train of neuron $j$ and $t_j^k$ are the times of its spikes. Note that such an autoencoder automatically maintains an accurate representation, because it responds to any encoding error larger than the firing threshold by increasing its response and in turn decreasing the error. It is also efficient, because neurons respond only when input and decoded signals differ. The autoencoder can be equivalently implemented by lateral connections, rather than feedback targeting the inputs (fig 1A). These lateral connections combine the feedforward connections and the decoding weights and they subtract from the feedforward inputs received by each neuron. The membrane potential dynamics in this recurrent network are described by:

$$\dot{\mathbf{V}} = -\lambda \mathbf{V} + \mathbf{Fs} + \mathbf{Wo} \qquad (1)$$

where $\mathbf{V}$ is the vector of the membrane potentials of the population, $\mathbf{s} = \dot{\mathbf{x}} + \lambda \mathbf{x}$ is the effective input to the population, $\mathbf{W} = -\mathbf{FD}$ is the connectivity matrix, and $\mathbf{o}$ is the population vector of the spike. Neuron $i$ has threshold $T_i = \|\mathbf{F}_i\|^2/2$ [15]. When input channels are independent and the feed-forward weights are distributed uniformly on a sphere then the optimal decoding weights $\mathbf{D}$ are equal to the encoding weights $\mathbf{F}$ and hence the optimal recurrent connectivity $\mathbf{W} = -\mathbf{FF}^T$ [17]. In the following we assume that this is always the case and we choose the feedforward weights accordingly.

In this auto-encoding scheme having a precise representation of the inputs is equivalent to maintaining a precise balance between excitation and inhibition. In fact, the membrane potential of a neuron is the projection of the global error of the network on the neurons's feedforward weight ($V_i = \mathbf{F}_i(\mathbf{x} - \hat{\mathbf{x}})$ [15]). If the output of the network matches the input, the recurrent term in the membrane potential, $\mathbf{F}_i\hat{\mathbf{x}}$, should precisely cancel the feedforward term $\mathbf{F}_i\mathbf{x}$. Therefore, in order to learn the connectivity matrix $\mathbf{W}$, we tackle the problem through balance, which is its physiological characterization. The learning rule that we derive achieves efficient coding by enforcing a precise balance at a single neuron level. The learning rule makes the network converge to a state where each presynaptic spike cancels the recent charge that was accumulated by the postsynaptic neuron (Fig 1B). This accumulation of charge is naturally represented by the postsynaptic membrane potential $V_i$, which jumps upon the arrival of a presynaptic spike by a magnitude given by the recurrent weight

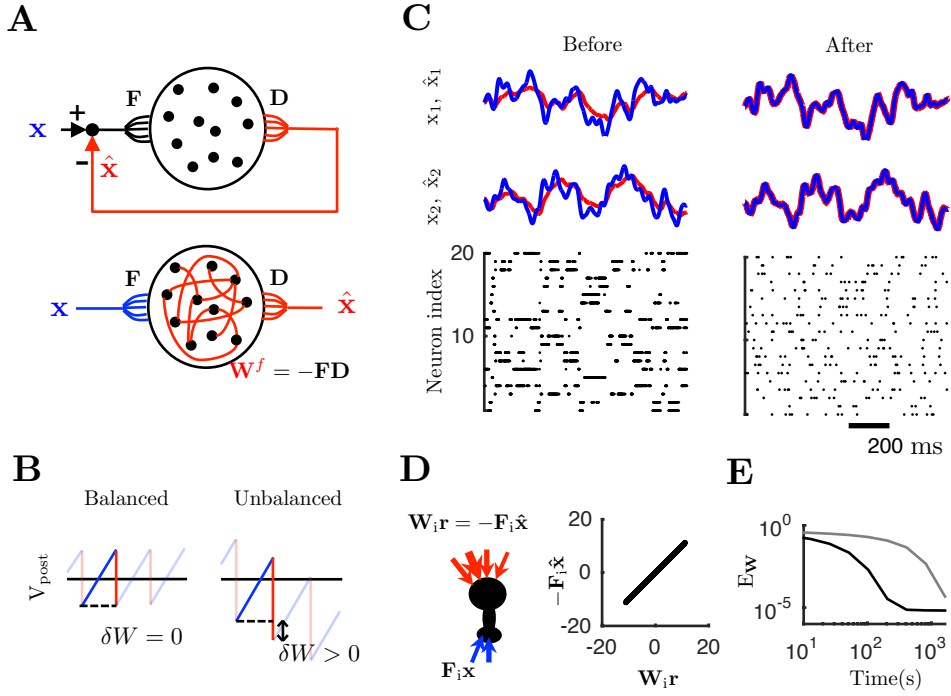

Figure 1: **A**: a network preforming predictive coding. Top panel: a set of unconnected leaky integrate-and-fire neurons receiving the error between a signal and their own decoded spike trains. Bottom panel: the previous architecture is equivalent to the recurrent network with lateral connections equal to the product of the encoding and the decoding weights. **B**: illustration of the learning of an inhibitory weight. The trace of the membrane potential of a postsynaptic neuron is shown in blue and red. The blue lines correspond to changes due to the integration of the feedforward input, and the red to changes caused by the integration of spikes from neurons in the population. The black line represents the resting potential of the neuron. In the right panel the presynaptic spike perfectly cancels the accumulated feedforward current during a cycle and therefore there is no learning. In the left panel the inhibitory weight is too strong and thus creates imbalance in the membrane potential; therefore, it is depressed by learning. **C**: learning in a 20 neuron network. Top panels: the two dimensions of the input (blue lines) and the output (red lines) before (left) and after (right) learning. Bottom panels: raster plots of the spikes in the population. **D**: left panel: after learning each neuron receives a local estimate of the output of the network through lateral connections (red arrows). right panel: scatter plot of the output of the network projected on the feedforward weights of the neurons versus the recurrent input they receive. **E**: the evolution of the mean error between the recurrent weights of the network and the optimal recurrent weights $-\mathbf{FF}^T$ using the rule defined by equation 2 (black line) and the rule in [16] (gray line). Note that our rule is different from [16] because it operates on a a finer time-scale and reaches the optimal balanced state with more than one order of magnitude faster. This speed-up is important because, as we will see below, some computations require a very fast restoration of this balance.

$W_{ij}$ due to the instantaneous nature of recurrent synapses. Because the two charges should cancel each other, the greedy learning rule is proportional to the sum of both quantities:

$$\delta W_{ij} \propto -(V_i + \beta W_{ij}) \tag{2}$$

where $V_i$ is the membrane potential of the postsynaptic neuron, $W_{ij}$ is the recurrent weight from neuron $j$ to neuron $i$, and the factor $\beta$ controls the overall magnitude of lateral weights and, therefore, the total spike count in the population. More importantly, $\beta$ regularizes the cost penalizing the total spike count in the population (i.e. $H(\mathbf{r}) = \mu \sum_i r_i$ where $\mu$ is the effective linear cost [15]). The example of an inhibitory synapse $W_{ij} < 0$ is illustrated in figure 1B. If neuron $i$ is too hyperpolarized upon the arrival of a presynaptic spike from neuron $j$, i.e., if the inhibitory weight $W_{ij}$ is smaller

than $-V_i/\beta$, the absolute weight of the synapse (the amplitude of the IPSP) is decreased. The opposite occurs if the membrane is too depolarized. The synaptic weights thus converge when the two quantities balance each other on average $W_{ij} = -\langle V_i \rangle_{t_j}/\beta$, where $t_j$ are the spike times of the presynaptic neuron $j$.

Fig 1C shows the learning in a 20-neuron network receiving random input signals. For illustration purposes the weights are initialized with very small values. Before learning, the lack of lateral connectivity causes neurons to fire synchronously and regularly. After learning, spike trains are sparse, irregular and asynchronous, despite the quasi absence of noise in the network. Even though the firing rates decrease globally, the quality of the input representation drastically improves over the course of learning. Moreover, the convergence of recurrent weights to their optimal values is typically quick and monotonic (Fig 1E).

By enforcing balance, the learning rule establishes an efficient and reliable communication between neurons. Because $\mathbf{V} = \mathbf{F}\mathbf{x} - \mathbf{F}\mathbf{F}^T\mathbf{r} = \mathbf{F}(\mathbf{x} - \hat{\mathbf{x}})$, every neuron has access - through its recurrent input - to the network's global coding error projected on its feedforward weight (Fig 1D). This local representation of network's the global performance is crucial in the supervised learning scheme we describe in the following sections.

## 2 Generating temporal dynamics within the network

While in the previous section we presented a novel rule that drives a spiking network into efficiently representing its inputs, we are generally interested in networks that perform more complex computations. It has been shown already that a network having two synaptic time scales can implement an arbitrary linear dynamical system [15]. We briefly summarize this approach in this section.

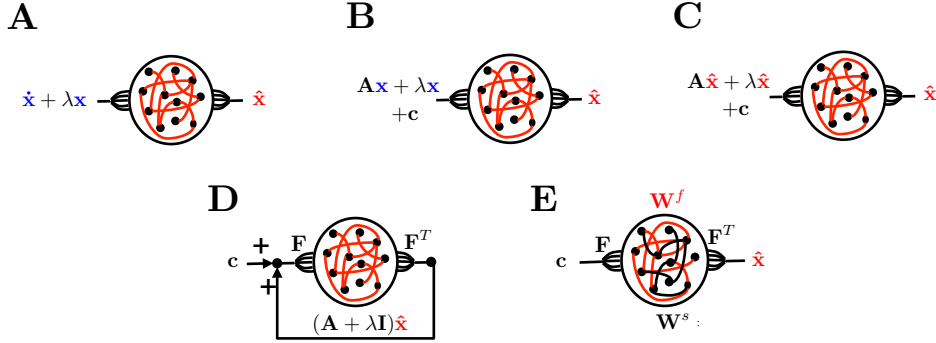

Figure 2: The construction of a recurrent network that implements a linear dynamical system.

In the autoencoder presented above, the effective input to the network is $\mathbf{s} = \dot{\mathbf{x}} + \lambda\mathbf{x}$ (Fig 2A). We assume that $\mathbf{x}$ follows linear dynamics $\dot{\mathbf{x}} = \mathbf{A}\mathbf{x} + \mathbf{c}$, where $\mathbf{A}$ is a constant matrix and $\mathbf{c}(t)$ is a time varying command. Thus, the input can be expanded to $\mathbf{s} = \mathbf{A}\mathbf{x} + \mathbf{c} + \lambda\mathbf{x} = (\mathbf{A} + \lambda\mathbf{I})\mathbf{x} + \mathbf{c}$ (Fig 2B). Because the output of the network $\hat{\mathbf{x}}$ approximates $\mathbf{x}$ very precisely, they can be interchanged. According to this self-consistency argument, the external input term $(\mathbf{A} + \lambda\mathbf{I})\mathbf{x}$ is replaced by $(\mathbf{A} + \lambda\mathbf{I})\hat{\mathbf{x}}$ which only depends on the activity of the network (Fig 2C). This replacement amounts to including a global loop that adds the term $(\mathbf{A} + \lambda\mathbf{I})\hat{\mathbf{x}}$ to the source input (Fig 2D). As in the autoencoder, this can be achieved using recurrent connections in the form of $\mathbf{F}(\mathbf{A} + \lambda\mathbf{I})\mathbf{F}^T$ (Fig 2E). Note that this recurrent input is the filtered spike train $\mathbf{r}$, not the raw spikes $\mathbf{o}$. As a result, these new connections have slower dynamics than the connections presented in the first section. This motivates us to characterize connections as fast and slow depending on their underlying dynamics. The dynamics of the membrane potentials are now described by:

$$\dot{\mathbf{V}} = -\lambda_V \mathbf{V} + \mathbf{F}\mathbf{c} + \mathbf{W}^{\mathbf{s}}\mathbf{r} + \mathbf{W}^{\mathbf{f}}\mathbf{o} \tag{3}$$

where $\lambda_V$ is the leak in the membrane potential, it is different from the leak in the decoder $\lambda$. It is clear from the previous construction that the slow connectivity $\mathbf{W}^{\mathbf{s}} = \mathbf{F}(\mathbf{A} + \lambda\mathbf{I})\mathbf{F}^T$, is involved

in generating the temporal dynamics of $\mathbf{x}$. Owing to the slow connections, the network is able to generate autonomously the temporal dynamics of the output and thus, only needs the command $\mathbf{c}$ as an external input. For example, if $\mathbf{A} = \mathbf{0}$ (i.e. the network implements a pure integrator), $\mathbf{W^s} = \lambda\mathbf{FF}^T$ compensates for the leak in the decoder by generating a positive feedback term that prevents the activity form decaying. On the other hand, the fast connectivity matrix $\mathbf{W^f} = -\mathbf{FF}^T$, trained with the unsupervised, voltage based rule presented previously, plays the same role as in the autoencoder; It insures that the global output and the global coding error of the network are available locally to each neuron.

## 3 Teaching the network to implement a desired dynamical system

Our aim is to develop a supervised learning scheme where a network learns to generate a desired output using an error feedback as well as a local learning rule. The learning rule targets the slow recurrent connections responsible for the generation of the temporal dynamics in the output, as seen in the previous section. Instead of deriving directly the learning rule for the recurrent connections, we first derive a learning rule for the matrix $\mathbf{A}$ of the linear dynamical system using simple results from control theory, and then we translate the learning to the recurrent network.

### 3.1 learning a linear dynamical system online

Consider the linear dynamical system $\dot{\hat{\mathbf{x}}} = \mathbf{M}\hat{\mathbf{x}} + \mathbf{c}$ where $\mathbf{M}$ is a matrix. We derive an online learning rule for the coefficients of the matrix $\mathbf{M}$, such that the output $\hat{\mathbf{x}}$ becomes after learning equal to the desired output $\mathbf{x}$. The latter undergoes the dynamics $\dot{\mathbf{x}} = \mathbf{Ax} + \mathbf{c}$. Therefore, we define $\mathbf{e} = \mathbf{x} - \hat{\mathbf{x}}$ as the error vector between the actual and the desired output. This error is fed to the mistuned system in order to correct and "guide" its behavior (Fig 3A). Thus, the dynamics of the system with this feedback are $\dot{\hat{\mathbf{x}}} = \mathbf{M}\hat{\mathbf{x}} + \mathbf{c} + K(\mathbf{x} - \hat{\mathbf{x}})$, where K is a scalar implementing the gain of the loop. The previous equation can be rewritten in the following form:

$$\dot{\hat{\mathbf{x}}} = (\mathbf{M} - K\mathbf{I})\hat{\mathbf{x}} + \mathbf{c} + K\mathbf{x} \tag{4}$$

where $\mathbf{I}$ is the identity matrix. If we assume that the spectra of the signals are bounded, it is straightforward to show, via a Laplace transform, that $\hat{\mathbf{x}} \to \mathbf{x}$ when $K \to +\infty$. The larger the gain of the feedback, the smaller the error. Intuitively, if K is large, very small errors are immediately detected and therefore, corrected by the system. Nevertheless our aim is not to correct the dynamical system forever, but to teach it to generate the desired output itself without the error feedback. Thus, the matrix $\mathbf{M}$ needs to be modified over time. To derive the learning rule for the matrix $\mathbf{M}$, we operate a gradient descent on the loss function $L = \mathbf{e}^T\mathbf{e} = \|\mathbf{x} - \hat{\mathbf{x}}\|^2$ with respect to the components of the matrix. The component $M_{ij}$ is updated proportionally to the gradient of $L$,

$$\delta M_{ij} = -\frac{\partial L}{\partial M_{ij}} = (\frac{\partial \hat{\mathbf{x}}}{\partial M_{ij}})^T \mathbf{e} \tag{5}$$

To evaluate the term $\partial \hat{\mathbf{x}}/\partial M_{ij}$, we solve the equation 4 in the simple case were inputs $\mathbf{c}$ are constant. If we assume that $K$ is much larger than the eigenvalues of $\mathbf{M}$, the gradient $\partial \hat{\mathbf{x}}/\partial M_{ij}$ is approximated by $\mathbf{E}_{ij}\hat{x}$, where $\mathbf{E}_{ij}$ is a matrix of zeros except for component $ij$ which is one. This leads to the very simple learning rule $\delta M_{ij} \approx \hat{x}_j e_i$, which we can write in matrix form as:

$$\delta\mathbf{M} \propto \mathbf{e}\hat{\mathbf{x}}^T \tag{6}$$

The learning rule is simply the outer product of the output and the error. To derive the learning rule we assume constant or slowly varying input. In practice, however, learning can be achieved also using fast varying inputs (Fig 3).

### 3.2 learning rule for the slow connections

In the previous section we derived a simple learning rule for the state matrix $\mathbf{M}$ of a linear dynamical system, driving it into a desired regime. We translate this learning scheme to the recurrent network described in section 2. To do this, two things have to be determined. First, we have to define the form of the error feedback in the recurrent network case. Second, we need to adapt the learning

rule of the matrix of the underlying dynamical system to the slow weights of the recurrent neural network.

In the previous learning scheme the error is fed to the dynamical system as an additional input. Since the input/decoding weight vector of a neuron $\mathbf{F}_i$ defines the direction that is relevant for its "action" space, the neuron should only receive the errors that are in this direction. Thus, the error vector is projected on the feedforward weights vector of a neuron before being fed to it. The feedback weights matrix is then simply equal to the feedforward weights matrix $\mathbf{F}$ (Fig 3A). Accordingly, equation 3 becomes:

$$\dot{\mathbf{V}} = -\lambda_V \mathbf{V} + \mathbf{Fc} + \mathbf{W^s r} + \mathbf{W^f o} + K\mathbf{Fe} \tag{7}$$

In the autoencoder, the membrane potential of a neuron represents the auto-coding error made by the entire network along the direction of the neuron's feedforward weights. With the addition of the dynamic error feedback and the slow connections, the membrane potentials now represent the error between obtained and desired network output trajectories.

To translate the learning rule of the dynamical system into a rule for the recurrent network, we assume that any modification of the recurrent weights directly reflects a modification in the underlying dynamical system. This is achieved if the updates $\delta\mathbf{W^s}$ of the slow connectivity matrix are in the form of $\mathbf{F}(\delta\mathbf{M})\mathbf{F}^T$. This ensures that the network always implements a linear dynamical system and guarantees that the analysis is consistent. The learning rule of the slow connections $\mathbf{W^s}$ is obtained by replacing $\delta\mathbf{M}$ by its expression according to equation 6 in $\mathbf{F}(\delta\mathbf{M})\mathbf{F}^T$:

$$\delta\mathbf{W^s} \propto (\mathbf{Fe})(\mathbf{F\hat{x}})^T \tag{8}$$

According to this learning rule, the weight update between two neurons, $\delta W_{ij}^s$, is proportional to the error feedback $\mathbf{F}_i\mathbf{e}$ received as a current by the postsynaptic neuron $i$ and to $\mathbf{F}_j\hat{\mathbf{x}}$, the output of the network projected on the feedforward weight of the presynaptic neuron $j$. The latter quantity is available to the presynaptic neuron through its inward fast recurrent connections, as shown for the autoencoder in Fig 1D.

One might object that the previous learning rule is not biologically plausible because it involves currents present separately in the pre- and post-synaptic neurons. Indeed, the presynaptic term may not be available to the synapse. However, as shown in the supplementary information of [15], the filtered spike train $r_j$ of the presynaptic neuron is approximately proportional to $\lfloor \mathbf{F}_j\hat{\mathbf{x}} \rfloor_+$, a rectified version of the presynaptic term in the previous learning rule. By replacing $\mathbf{F}_j\hat{\mathbf{x}}$ by $r_j$ in the equation 8 we obtain the following biologically plausible learning rule:

$$\delta W_{ij}^s = E_i r_j \tag{9}$$

Where $E_i = \mathbf{F}_i\mathbf{e}$ is the total error current received by the postsynaptic neuron.

### 3.3 Learning the underlying dynamical system while maintaining balance

For the previous analysis to hold, the fast connectivity $\mathbf{W}^f$ should be learned simultaneously with the slow connections using the learning rule defined by equation 2. As shown in the first section, the learning of the fast connections establishes a detailed balance on the level of the neuron and guarantees that the output of the network is available to each neuron through the term $\mathbf{F}_j\hat{\mathbf{x}}$. The latter is the presynaptic term in the learning rule of equation 8. Despite not being involved in the dynamics per se, these fast connections are crucial in order to learn any temporal dynamics. In other words, learning a detailed balance is a pre-requirement to learn dynamics with local plasticity rules in a spiking network. The plasticity of the fast connections restores very quickly any perturbation to the balance caused by the learning of the slow connections.

### 3.4 Simulation

As a toy example, we simulated a 20-neuron network learning a 2D-damped oscillator using a feedback gain $K = 100$. The network is initialized with weak fast connections and weak slow connections. The learning is driven by smoothed gaussian noise as the command $\mathbf{c}$. Note that in the initial state, because of the absence of fast recurrent connections, the output of the network does not depend linearly on the input because membrane potentials are hyperpolarized (Fig 3B). The network's output is quickly linearized through the learning of the fast connections (equation 2 by enforcing a

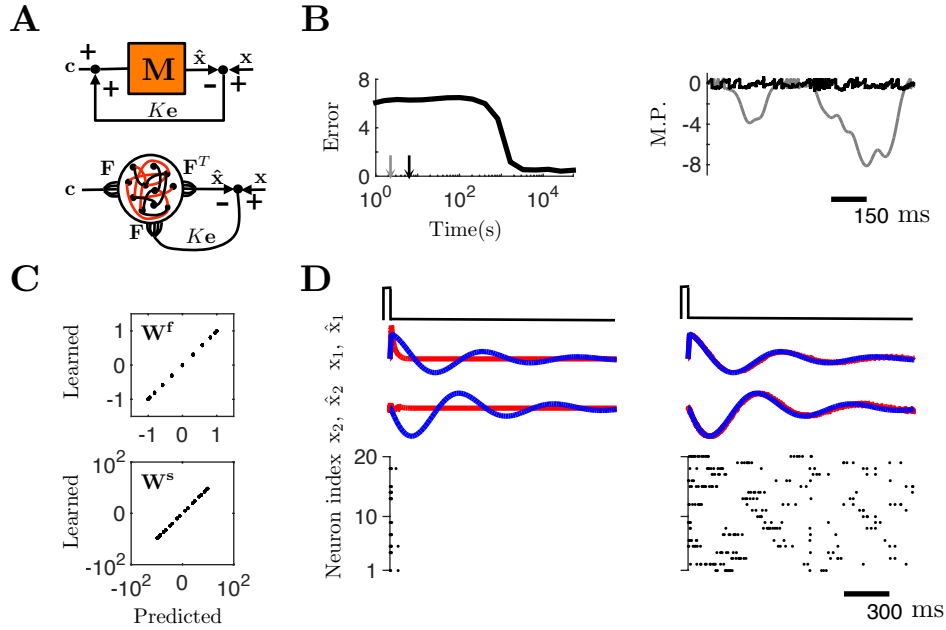

Figure 3: Learning temporal dynamics in a recurrent network. **A**, Top panel: the linear dynamical system characterized by the state matrix **M**, receives feedback signaling the difference between its actual output and a desired output. Bottom panel: a recurrent network displaying slow and fast connections is equivalent to the top architecture if the error feedback is fed into the network through the feedforward matrix **F**. **B**: a 20 neuron network learns using equations 9 and 2. Left panel: the evolution of the error between the desired and the actual output during learning. The black and grey arrows represent instances where the time course of the membrane potential is shown in the next plot. Right panel: the time course of the membrane potential of one neuron at two different instances during learning. The gray line corresponds to the initial state while the black line is a few iterations after. **C**: scatter plots of the learned versus the predicted weights at the end of learning for fast (top panel) and slow (bottom panel) connections. **D**, top panels: the output of the network (red) and the desired output (blue), before (left) and after (right) learning. The black solid line on the top shows the impulse command that drives the network. Bottom panels: raster plots before and after learning. In the left raster plot there is no spiking activity after the first 50 ms.

balance on the membrane potential (Fig 3B): initial membrane potentials exhibit large fluctuations which reduce drastically after a few iterations (Fig 3B). On a slower time scale the slow connections learn to minimize the prediction error using the learning rule of equation 9. The error between the output of the network and the desired output decreases drastically (Fig 3B). To compute this error, different instances of the connectivity matrices were sampled during learning. The network was then re-simulated using these instances while fixing K=0 in oder to mesure the performance in the absence of feedback. At the end of learning the slow and fast connections converge to their predicted values $\mathbf{W^s} = \mathbf{F}(\mathbf{A} + \lambda\mathbf{I})\mathbf{F}^T$ and $\mathbf{W^f} = -\mathbf{F}\mathbf{F}^T$ (Fig 3C). The presence of the feedback is no longer required for the network to have the right dynamics (i.e. we set $K = 0$ and obtain the desired output (Fig 3D and 3B). The output of the network is very accurate (representing the state **x** with a precision of the order of the contribution of a single spike), parsimonious (i.e. it does not spend more spikes than needed to represent the dynamical state with this level of accuracy) and the spike trains are asynchronous and irregular. Note that because the slow connections are very weak in the initial state, spiking activity decays quickly after the end of the command impulse due to the absence of slow recurrent excitation (Fig 3D).

**Simulation parameters** Figure 1 : $\lambda = 0.05$, $\beta = 0.51$, learning rate: 0.01. Figure 3 : $\lambda = 50$, $\lambda_V = 1$, $\beta = 0.52$, $K = 100$, learning rate of the fast connections: 0.03, learning rate of the slow connections: 0.15.

# 4 Discussion

Using a top-down approach we derived a pair of spike-based and current-based plasticity rules that enable precise supervised learning in a recurrent network of LIF neurons. The essence of this approach is that every neuron is a precise computational unit that represents the network error in a subspace of dimension 1 in the the output space. The precise and distributed nature of this code allows the derivation of local learning rules from global objectives.

To compute collectively, the neurons need to communicate to each other about their contributions to the output of the network. The fast connections are trained in an unsupervised fashion using a spike-based rule to optimize this communication. It establishes this efficient communication by enforcing a detailed balance between excitation and inhibition. The slow connections however are trained to minimize the error between the actual output of the network and a target dynamical system. They produce currents with long temporal correlations implementing the temporal dynamics of the underlying linear dynamical system. The plasticity rule for the slow connections is simply proportional to an error feedback injected as a current in the postsynaptic neuron, and to a quantity akin to the firing rate of the presynaptic neuron. To guide the behavior of the network during learning, the error feedback must be strong and specific. Such strength and specialization is in agreement with data on climbing fibers in the cerebellum [18–20], who are believed to bring information about errors during motor learning [21]. However, in this model, the specificity of the error signals are defined by a weight matrix through which the errors are fed to the neurons. Learning these weights is still under investigation. We believe that they could be learned using a covariance-based rule.

Our approach is substantially different form usual supervised learning paradigms in spiking networks since it does not target the spike times explicitly. However, observing spike times may be misleading since there are many combinations that can produce the same output [15, 16]. Thus, in this framework, variability in spiking is not a lack of precision but is the consequence of the redundancy in the representation. Neurons having similar decoding weights may have their spike times interchanged while the global representation is conserved. What is important is the cooperation between the neurons and the precise spike timing relative to the population. For example, using independent poisson neurons with instantaneous firing rates identical to the predictive coding network drastically degrades the quality of the representaion [15].

Our approach is also different from liquid computing in the sense that the network is small, structured, and fires only when needed. In addition, in these studies the feedback error used in the learning rule has no clear physiological correlate, while here it is concretely injected as a current in the neurons. This current is used simultaneously to drive the learning rule and to guide the dynamics of the neuron in the short term. However, it is still unclear what the mechanisms are that could implement such a current dependent learning rule in biological neurons.

An obvious limitation of our framework is that it is currently restricted to linear dynamical systems. One possibility to overcome this limitation would be to introduce non-linearities in the decoder, which would translate into specific non-linearities and structures in the dendrites. A similar strategy has been employed recently to combine the approach of predictive coding and FORCE learning [7] using two compartment LIF neurons [22]. We are currently exploring less constraining forms of synaptic non-linearities, with the ultimate goal of being able to learn arbitrary dynamics in spiking networks using purely local plasticity rules.

**Acknowledgments**

This work was supported by ANR-10-LABX-0087 IEC, ANR-10-IDEX-0001-02 PSL, ERC grant FP7-PREDISPIKE and the James McDonnell Foundation Award - Human Cognition.

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
