[Reviews · NeurIPS 2015]

Submitted by Assigned_Reviewer_1

The authors present a method for learning to cancel input currents in a spiking network, which they then use to learn linear dynamical systems. The mathematics looks correct, and the paper is written a fairly clear manner. I know the related literature fairly well, and it appears original. The learning of linear dynamical systems in the paper is fairly limited, but the model constitutes a step in the right direction. I am impressed by how small the networks are that learn these dynamics. Typical spiking network learning uses huge numbers of neurons to approximate effective population rates. Not so here. It's neat.
Summary: It's hard to make spiking networks learn interesting dynamics, and the authors present a method for doing so that is only somewhat unnatural. I think this work is worthy of acceptance.

Submitted by Assigned_Reviewer_2

Summary:

This paper introduces a new learning framework in leaky integrate and fire neurons, which permits a recurrent network to efficiently learn linear dynamical systems. The approach uses weight changes at two timescales: fast weight changes quickly balance excitation and inhibition, while slower weight changes learn the structure of the LDS. A key insight is that the fast plasticity which balances excitation and inhibition distributes a global signal about the network's performance to all neurons, enabling error driven learning of the LDS with a local learning rule.

Major comments:

This paper presents the intriguing idea of using the balance of excitation and inhibition to distribute global error information throughout a neural network, permitting supervised learning with a local learning rule.

Moreover, the scheme introduced is based on predictive coding, which as the paper shows, naturally leads to sparse irregular spiking activity. On this subtle view, neural firing in response to an identical input will not yield identical precise spike times; but the particular spike times for each input presentation are nonetheless precisely arranged, and cannot be replaced by a rate coded approximation without a drop in fidelity or efficiency.

The theoretical arguments of the paper are lucid and clearly expressed, and contain many smaller insights such as how various signals can be incorporated into recurrent weights, or swapped to decoded rather than actual outputs. A useful insight is the idea that a presynaptic term necessary for error-driven learning may be estimated by filtered spikes arriving at the postsynaptic neuron's synapse. Overall this paper shines in its details.

The toy examples given in simulation are sufficient to demonstrate that the technique does indeed work as expected. The paper would be stronger with a more compelling application, but already makes an interesting contribution.

It would be useful to add details of the simulations, perhaps as supplementary materials, to allow for replications. What are the values of \lambda, \mu, \lambda_V, etc?

There are several limitations to the method at present. Most notably, only linear dynamical systems can be represented. Each neuron requires a strong, precisely targeted error feedback signal. It is only the recurrent weights that learn, with the feedforward and feedback weights being fixed by hand. The network and feedback signal must operate directly on the state of the LDS, rather than an output derived from the LDS (e.g. y = Cx where C is a new matrix). The network is closely tied to the task of representing its input accurately, but in general it is not input reconstruction but a transformation from input to output that must be learned. These challenges are reasonable to leave for future work.

Minor comments:

ln 104: "to the its" -> "to its" ln 175: \hat x should not be transposed, I believe ln 215: "for these reason" -> "for this reason" Fig 3B caption ln 351: "top/bottom panel"-> left/right panel ln 351: "using the equations" -> "using equations" ln 398: "who are believed" -> "which are believed"
Summary: This paper presents the intriguing idea of using the balance of excitation and inhibition to distribute global error information throughout a spiking neural network, permitting supervised learning with a local learning rule. This is a highly novel role for balanced excitation/inhibition that many will find interesting.

Submitted by Assigned_Reviewer_3

I greatly appreciate the problem the authors are trying to solve: supervised learning is difficult to implement in a biologically plausible manner, indeed known algorithms (such as backprop through time) are typically non-local. The authors propose a neat solution: unsupervised learning of fast recurrent connections establishes a detailed balance of E and I currents, which is then shown to make relevant aspects of the global output of the network available to each neuron locally. These global-made-local signals can then be used to learn slower connections in order to implement a linear dynamical system.

I find the idea really appealing, but I think the results are rather preliminary and the whole study could have been more thorough.

- while the spike-based rule used to learn a detailed balance can certainly be shown to achieve its goal in an isolated postsynaptic neuron, it is by no means guaranteed to work in a recurrent network (for just about the same reasons that make local approximations to error backpropagation close to useless in a recurrent net). Why is it that it works here?

- learning the slow connections requires very strong feedback (K large). The authors claim that after learning, since the error has vanished anyways, K could be set to zero without affecting performance. Is that true?

Is there a figure where we can see that? If the error was exactly zero, then I would buy that, but since it's never perfect, one is allowed to wonder about the stability of the K=0 solution to small errors. Indeed the error can be reduced arbitrarily by making K very large, without even touching the slow weights, so clearly in that case a very small error isn't at all predictive of successful outputs with K=0; for a stable solution to emerge with K small, the error with K large should be further reduced (through learning) by an amount that ought to be made more precise mathematically. I guess what I'm uncomfortable with is that if the authors' claim was true, we could imagine decreasing K progressively as W_slow is being learned, but then at some point we would hit the limit of validity of the first-order expansion of (I-M/K)^{-1} used to derive Eq7 and run into trouble. Am I missing something here?

- somehow, the authors postpone the difficult problem of local supervised learning to the (in my opinion, not much easier) learning of identical input, output, and error feedback weights that must all be equal to the same matrix F. Such learning is not addressed here. Could the authors discuss/comment on that?

- in their derivations, the authors assume that the inputs to the dynamical system they are trying to learn are constant (or very slow changing). What happens if inputs are faster? (btw, what was the author's choice of input c in the learning setup of Fig3?)

Some minor comments/typos:

- line 94: what does it mean for the feedforward weights to "evenly span the input space"?

- isn't there a sign issue in Eq2? If W_ij > 0 and V_i > 0, the learning rule is going to strengthen W_ij which makes the imbalance even worse.

- l175: adjust to bold, "vector font"

- trying to derive Eq.7, I instead got \delta M_{ij} \approx (c_j + Kx_j) e_i. Did the authors drop the constant term on the basis of K being large?

- the whole paragraph l.278-l.285 was rather obscure to me; in the end, don't the authors want the error to be fed back through F because of the Fe term in Eq. 9? Couldn't that be stated more simply/clearly?

- l221: different from - ref list Gutig - Guetig of Guetig - ref list Deneve - Deneve - caption of fig3B - replace top/bottom by left/right - l387 the the

- l403 different from

UPDATE following author's responses: I'm updating my evaluation to 8 due to a satisfying response to my second comment in particular.
Summary: This paper provides a solution to the problem of getting biologically implementable gradients for supervised learning in recurrent networks. I found the author's solution -- based on a detailed E/I balance -- very neat, but I've been disappointed by the lack of rigor in the exposition and the preliminary nature of the numerical results.

Submitted by Assigned_Reviewer_4

- In general, I had difficulty reconciling the text and Eq. 2. In particular, these two sentences are not clear at all, and should be revised carefully: (pg 2, line 103) Since we want to keep ... (pg 3, line 159) The factor 1/2 imposes ... - In the discussion, the authors speculate on extending to nonlinear systems, however the extension to time-varying linear systems (e.g., A_t) is not discussed. - The equivalence of the two circuits in Fig. 1A is not in a strict circuit-theoretic sense and assumes asymptotic behavior. The text and the figure should reflect this distinction. - (pg 5, line 250) The statement using Laplace transform should assume that the frequency is bounded. - There are quite a few typos. Please check the manuscript carefully.
Summary: This paper proposes a learning algorithm in spiking recurrent networks. The paper demonstrates the algorithm's capability of learning stable linear time-invariant dynamical systems. Three features of the appraoch are (i) the locality of the learning rules, (ii) the maintenance and utilization of balance in network activity, and (iii) the utilization of two different time-scales within the network to facilitate learning.

Overall, I think the paper provides a welcome advance to learning in spiking neural networks although the class of learnable dynamical systems with the proposed model is somewhat narrow. The presentation is not clear especially in Section 1, and includes vague sentences at critical places and minor technical mistakes.

Author Feedback
Author rebuttal: We would like to thank the reviewers for their thoughtful comments and their constructive criticism. One frequently raised issue is the simplicity of the toy example that illustrates the learning in Fig 3. We will remediate this by simulating a 4 D system representing a planar robot arm and show how the tuning curves to movement direction develop as a result of learning. Another issue is the learning of the feed forward weights and the feedback weights that are assumed to be equal to each other here. Currently, we have a rule that permits the learning of the optimal feed forward weight. Nonetheless we do not have yet a learning rule for the feedback weights. We are investigating this issue and we think that those weights could be learned with a similar rule to that of the feed forward weight. We did not show the learning of the latter here because we thought that it would be of little direct interest to our problem without a rule for the feedback weights. Finally we will rewrite the paragraphs and sections that the reviewers marked as unclear, and we will report the parameters used in the simulations.

#Assigned_Reviewer_1

We hope that we responded to your concerns in the general comments above.

#Assigned_Reviewer_2

Indeed there are no absolute mathematical guarantees for the fast rule to converge in a recurrent network. However, this rule is an approximation of another exact, but biologically implausible rule that is not shown in the paper. It can be shown that the only fixed points of the latter are FF^T. Although we cannot show that they are stable, we do observe in numerical simulations that the fast weights always converge to this solution using random initializations. One condition for the learning to work is that inputs must be independent across input channels.

It is not stated clearly in the text, but Fig 3B, left, shows the reconstruction error when K=0. More precisely, the learning was achieved using a large fixed K. Different instances of the connectivity matrices were sampled during learning. The network was then re-simulated using these instances while fixing K=0 in oder to mesure the performance in the absence of feedback. It is clear from the plot that at the end of learning the performance is very good. This is the case because the slow lateral connections are learned very accurately (Fig 3C). In fact, at the end of learning, the currents provided by the feedback are two orders of magnitude smaller than the currents provided by the slow connections and therefore have very little influence.

The rule is derived assuming slowly changing input in order to make the derivations possible to do analytically. However, in practice the inputs used can be very fast. For instance, in Fig 3 the input is a filtered Gaussian noise.

#Assigned_Reviewer_3

We only considered the extension to non-linear systems because non-linear dynamics are frequently used in order to model many cognitive tasks, e.g decision making, memory retrieval, etc.

The two circuits are strictly equivalent if in the top panel of Fig 1A, the leaky integration takes place at the node that is on the left of the population.

#Assigned_Reviewer_6

It is stated that our work is simply a spiking version of an earlier work by Gomi and Kawato (1993, perhaps?) and therefore it is of limited interest. We strongly disagree with this claim for three main reasons.

First, translating a learning scheme for rate neurons into a learning scheme for spiking neurons is usually very challenging. Indeed, rate neurons and spiking neurons are not always equivalent in a phenomenological sense (Brunel 2000, Ostojic 2014). An example is that of FORCE learning (Sussillo and abbott 2009) which is very hard to translate into a spiking setup.

Second, our approach is substantially different and it is not an extension of the cited work. It uses recurrent networks while the cited paper assumes feed forward architectures, such as a multilayered perceptron. In addition to its large presence in the brain, recurrence in our case permits cooperation between neurons and makes the neural code very efficient (few spikes and very high accuracy). Moreover, the learning rule in the cited work is a priori non-local.

Third and most importantly, we link widely observed phenomena in the brain such as balance between excitation and inhibition and irregular firing, to neural coding and learning.